Polyester or epoxy: assessing embedding product efficacy in paleohistological methods

Heck Christian T. ctheck@okstate.edu
Volkmann Gwyneth
Woodward Holly N.
Department of Biomedical Sciences, Oklahoma State University Center for Health Sciences , Tulsa , OK , United States of America
Hedrick Brandon
Electronic publication date: 2020 Dec 15
Publication date: 2020
Volume: 8
Electronic Location ID: e10495
Received 2020 Aug 10; Accepted 2020 Nov 14
Copyright: ©2020 Heck et al.
Copyright year: 2020
Copyright holder: Heck et al.
License: This is an open access article distributed under the terms of the Creative Commons Attribution License, which permits unrestricted use, distribution, reproduction and adaptation in any medium and for any purpose provided that it is properly attributed. For attribution, the original author(s), title, publication source (PeerJ) and either DOI or URL of the article must be cited.
License URL: https://creativecommons.org/licenses/by/4.0/

Keywords: Paleohistology, Polyester, Epoxy, Osteohistology, Histotechniques, Bone microstructure, Embedding, Paleontology

Funding: The authors received no funding for this work.

==============================
Histological examination of bone microstructure provides insight into extant and extinct vertebrate physiology. Fossil specimens sampled for histological examination are typically first embedded in an inexpensive polyester resin and then cut into thin sections, mounted on slides, and polished for viewing. Modern undecalcified bone is chemically processed prior to embedding in plastic resin, sectioning, mounting, and polishing. Conversely, small fossil material and modern undecalcified bone are typically embedded in higher priced epoxy resin because these specimen types require final sections near or below 100 µm thick. Anecdotal evidence suggests thin sections made of polyester resin embedded material polished thinner than 100 µm increases likelihood of sample peeling, material loss, and is unsuitable for modern tissue and small fossil material. To test this assertion, a sample of modern bones and fossil bones, teeth, and scales were embedded in either polyester resin or epoxy resin. Embedded specimens were sectioned and mounted following standard published protocol. Thin sections were ground on a lapidary wheel using decreasing grit sizes until tissue microstructure was completely discernible when viewed under a polarizing light microscope. Additionally, eight prepared thin sections (four from polyester resin embedded specimens and four from epoxy resin embedded specimens) were continuously ground on a lapidary wheel using 600 grit carbide paper until peeling occurred or material integrity was lost. Slide thickness when peeling occurred was measured for comparing slide thickness when specimen integrity was lost between the two resin types. Final slide thickness ranged from 38 µm to 247 µm when tissue was identifiable using a polarizing microscope. Finished slide thickness varied between resin types despite similar tissue visibility. However, finished slide thickness appears more dependent on hard tissue composition than resin type. Additionally, we did not find a difference of slide thickness when material was lost between resin types. The results of this preliminary study suggest that polyester resins can be used for embedding undecalcified modern hard tissues and fossilized hard tissues without loss of tissue visibility or material integrity, at least in the short term.

Introduction

Histological examination of bone allows interpretation of relative growth rates, absolute age, pathologies, and life history reconstructions of extinct and extant taxa (Marín-Moratalla, Jordana & Köhler, 2013; Cubo et al., 2015; Woodward et al., 2015; Calderon et al., 2019). Vertebrate paleontology studies increasingly incorporate osteohistology, or bone histotechniques, for these reasons, with large-sample studies becoming more common. The recognized utility of osteohistology necessitates investigating the cost-effectiveness of consumables involved to reduce expense, especially concerning large sample sizes.

Decalcification of modern hard tissues is necessary for specific staining protocol at the sacrifice of the mineral component (Skinner, 2003). However, preparation of hard tissues without decalcification allows for investigation of mineralization patterns and direct comparisons with fossil specimens (Scarano, Orsini & Piattelli, 2003; Skinner, 2003; Straehl et al., 2013), as the unmineralized component of bone typically degrades prior to fossilization. Methodology for the preparation of undecalcified hard tissues, also simply termed ‘calcified tissue’, for histological examination varies but generally includes stepwise tissue fixation, dehydration, and clearing prior to specimen embedding in resin, mounting the embedded specimen to a slide with glue, and thin section polishing (Fig. 1) (An et al., 2003; Schweitzer et al., 2007). Soft tissues in bone, e.g., oils, fats, and the collagen component of bone, are lost during the process of fossilization, thus, preparation of fossil hard tissues does not include the initial chemical treatments of modern hard tissues but still requires embedding in resin for stabilization, mounting, and polishing (Fig. 1) (Chinsamy & Raath, 1992; Wilson, 1994; Lamm, 2013). Embedding, or investing, biological material for histological study was first introduced by Klebs (1869) using paraffin as the embedding medium (Sanderson et al., 1988). Over the next century advancements in petrography and biological histology developed, and new methodologies for processing specimens emerged; however, protocol for histological processing of mineralized fossil material in publications was often absent or vague. Mineralized osteohistological studies have since utilized a variety of embedding mediums, with epoxy and polyester resins being common for extant bone and fossil bone respectively. However, choice of fossil and modern hard tissue embedding medium does not appear established on the basis of resin efficacy. Instead, resin selection seems to be personal preference or based off of published corporate technical notes (Ahmed & Vander Voort, 2000; Ahmed & Vander Voort, 2003). For instance, Chinsamy & Raath (1992) were the first to publish a detailed protocol for their preparation of fossil bone for histological study and utilized epoxy resin for their embedding medium. They state, “…any resin or other rigid, clear mounting medium which does not interfere with the structure or optical properties of the tissues could be used” (Chinsamy & Raath, 1992, p. 40). Wilson (1994) also details methods used for preparing fossil bone for histological analysis and lists polyester resin as the preferred embedding medium for fossil bone. Lamm (2013) thoroughly describes the methodology developed at the Museum of the Rockies (Bozeman, MT, USA) for preparation and sectioning of fossil specimens for histological sampling. Lamm (2013) states that small fossils between one millimeter and one centimeter in length benefit from epoxy resin embedding due to the low viscosity of epoxy, which increases resin penetration, but that polyester resin is suitable for larger fossil material.

Figure 1 Simplified protocol for osteohistological protocol.

In standard protocol, processing of modern specimens requires chemical processing prior to embedding in epoxy resin. Fossil samples do not require chemical processing, but consolidants must be removed prior to embedding in polyester resin.

Resin choice is further compounded by price, with some epoxy resins costing up to 479% that of some polyester resins at the time of this publication. Such expenses can be prohibitive for underfunded researchers and institutions and necessitates determining cost effective alternatives. Here, we investigate the efficacy of polyester and epoxy embedding mediums (of different price points) commonly used in histological studies of fossil bone and modern undecalcified hard tissues to determine the variables requiring higher priced epoxy resins in the event that polyester and epoxy resins are similar in functionality.

Materials & Methods

Fossil and modern hard tissues were chosen for sampling to test the efficacy of polyester and epoxy resins. The fossils are donated bones, teeth, and fish scales of unknown provenience. Fossil specimens include a turtle femur, two ornithischian dinosaur teeth, gar scale, crocodile scute, and a rib and bone fragment of unknown taxa. Modern bones were either purchased raw from a local grocery store (domestic chicken (Gallus gallus domesticus)) or collected as salvage (nine-banded armadillo (Dasypus novemcinctus)). Salvage was collected under Oklahoma collecting permits. The chicken humerus, tibia, and femur were sampled as well as both calcanea from the nine-banded armadillo.

Fossil material was thoroughly scrubbed with an acetone-soaked brush to remove any consolidants from the bone surface. Intensive exposure to acetone can have deleterious effects on fossil bone, but we did not observe any damage to fossils from acetone washing. Specimens were placed in small silicone containers and vacuum impregnated with either Silmar-41 two-part polyester resin, a commonly manufactured polyester resin, or Buehler Epothin (1 or 2, see Table 1) two-part epoxy resin (Buehler Ltd.). A variety of epoxy and polyester resins are utilized by osteohistological studies, but the two resins tested here are commonly used in paleohistological embedding (Lamm, 2013). Buehler Epothin 1 became unavailable mid-way through the experiment and was replaced by two-part Buehler Epothin 2. We assume there is no major difference in efficacy between Epothin 1 and Epothin 2. Specimen processing then followed standard protocol outlined in Lamm (2013).

Table 1 Material histologically sampled using either polyester resin or epoxy resin as the embedding medium.

Resin type	Specimen age	Specimen	Element	No. of sections	
Polyester Resin	Modern	Domestic Chicken	Humerus	2	
(Silmar-41)			Femur	2	
			Tibia	2	
		Nine-banded Armadillo	Right Calcaneum*	2	
	Fossil	Indet. Turtle	Femur*	2	
		Gar	Scale	2	
		Indet. Crocodile	Scute	2	
		Indet. Ornithopod	Tooth*	2	
		Unknown	Rib	2	
		Unknown	Fragment*	1	
Epoxy Resin	Modern	Nine-banded Armadillo	Left Calcaneum*	2	
(Epothin-1)	Fossil	Indet. Turtle	Femur*	1	
		Unknown	Rib	1	
		Indet. Ceratopsian	Tooth*	2	
(Epothin-2)		Unknown	Fragment*	1	
Notes.

Specimens sampled for this study included modern and fossil hard tissues.

* indicates material used to compare specimen integrity during thin section grinding and polishing,

All modern bones were processed prior to embedding using modified techniques from An et al. (2003) and Schweitzer et al. (2007) and outlined here. Modern material was soaked in warm water mixed with 1% Tergazyme Enzyme Detergent (Alconox Inc.) to degrade and to ease the removal of soft tissues from the bones. Specimens were air dried and remaining connective tissues and muscle remnants were removed via dissection. Bones were fixed in 10% formalin solution for 2-3 days. Specimens were then dehydrated in step-wise increasing concentrations of ethanol starting at 70% EtOH for 48 h, followed by 85% EtOH for 48 h, and finishing in 100% EtOH for 48 h. Specimens were then cleared in Clear-Advantage Xylene Substitute (Polysciences Inc.) for 2–4 h and set aside until dry (24–48 h under a fume hood). Drying specimens after clearing introduces air back into the bone structure, but vacuum embedding replaces the reintroduced air with the embedding resin. Embedding procedure then proceeded as described above for fossil material (see Table 1 for embedding resin type used for each specimen).

One to two thin sections were generated from each embedded specimen (Table 1). A Buehler Isomet 1000 saw (Buehler Ltd.), equipped with a 6″diamond cutoff blade, was used to cut thick wafers of approximately 2.5 mm from each embedded specimen block. One side of each wafer was ground on a Buehler Ecomet 4 lapidary grinder/polisher (Buehler Ltd.) using silicon carbide paper from 600 grit to 800 grit. Additionally, one side of the plastic slides was “frosted” using 600 grit silicon carbide paper on the Ecomet 4. Frosting of the wafer and plastic slide permits better adherence when glue is applied. Using 600 grit silicon carbide paper does not create scratches large enough to affect visibility of the finished section with microscope viewing. Wafers were placed under a fume hood for 24 h to dry. After drying, wafers were mounted to frosted plastic slides with Starbond cyanoacrylate glue of medium viscocity to form a thin-section slide. Lamm (2013) recommends that polyester embedded specimens be mounted to glass slides using two-part, two-ton epoxy, while epoxy embedded specimens be mounted to plastic slides using cyanoacrylate glue for better adherence. Recent processing of thin sections on glass with two-ton epoxy as the adhesive resulted in artifacts at the microscopic level. Although not visible in plane light, the artifacts appear as tiny birefringent square flakes in cross polarized light (Fig. 2). The presence of birefringent flakes is not isolated to any single brand of two-part epoxy, and are only present in the hardener component (H. Woodward, 2019, pers. obs.). However, the artifact can be eliminated by reheating the hardener component to 50 °C (HNW pers. obs.). Here, we use plastic slides for all specimens because of the much lower cost of plastic slides relative to glass, and apply cyanoacrylate glue to (1) avoid potential visual complications caused by the use of two-ton epoxy and (2) continue following procedure outlined in Lamm (2013). Thin-sections (wafers mounted on slides) were set under a fume hood for 24 h to cure and were then removed and allowed to cure for an additional 24 h. Thin-sections were ground and polished using silicon carbide paper of decreasing grit sizes beginning at 320 grit and ending with 800 grit on a lapidary wheel until bone microstructure was visible and identifiable under a polarizing light microscope. Thin-section grinding on the lapidary wheel was controlled by hand and, thus, thin-sections were subjected to slight pressure variation during the grinding process. Slide holders can be used to eliminate pressure variation during lapidary wheel grinding, however, grinding was hand controlled in this study to better simulate low cost thin-section preparation techniques. Thin-sections were further polished by hand with 5 µm and 1 µm solutions. Osteohistological studies often rely on qualitative descriptions such as that of bone tissue organization and vascular canal organization. Therefore, clarity of tissue organization and visibility between specimens embedded in the two resin types were qualitatively assessed by the authors. Thickness of finished slides was averaged for each specimen, but a targeted thickness was not set due to differences in transparency of tissue organizations. Differences in resin refractive index were not taken into account when assessing tissue visibility.

Figure 2 Visual obstructions in the mounting medium 2-ton epoxy resin.

The 2-ton epoxy resin is used as a mounting medium for polyester resin embedded specimens to glass slides. (A) A drop of 2-ton epoxy resin imaged showing ’confetti’ visual obstructions. (B) ”Confetti” obstructing tissue visibility in a Maiasaura tibia cross-section. Both images taken with a camera mounted to a polarizing light microscope with a 1/4 lambda wave plate.

Slide thickness and specimen peeling

Two general kinds of damage can occur during grinding and polishing of hard tissues: (1) hard tissue material tearing, or popping, off the slide and (2) complete removal of specimen tissue due to excessive polishing. Lamm (2013) suggests epoxy resin performs better with material requiring extremely low thickness for tissue visibility whereas similar material embedded in polyester resin may succumb to the second type of damage during the grinding and polishing stage. Four specimens used in the study were chosen for further testing the ability of each resin to maintain specimen integrity when polished aggressively. The fossil rib and unknown bone fragment were carefully broken in half using a small hammer, and one half was embedded in Silmar-41 polyester resin and the other half embedded in either Buehler Epothin 1 or Epothin 2 epoxy resin (Table 1). Embedding protocol followed protocol previously stated. Testing each half in a different resin eliminated potential variation in tissue reaction based on mineral density, bone tissue organization, and/or vascular density. Similarly, one fossil tooth and the right nine-banded armadillo calcaneum was embedded in Silmar-41 while a second fossil tooth and the left nine-banded armadillo calcaneum was embedded in Buehler Epothin 2. Specimen processing and thin section preparation followed the previously stated protocol and the resultant thin section slides were polished on a lapidary wheel with 600 grit carbide paper until light could pass through the specimen. Thin sections were then polished further on the lapidary wheel using 800 grit carbide paper until specimen integrity was lost (damage type (1) or (2) as defined previously). Thickness of thin sections at moment of lost integrity was measured using a digital micrometer. Resultant thicknesses were compared between specimens embedded in each resin.

Results

We qualitatively assessed thin-sections produced from specimens embedded in the two resin types, Silmar-41 polyester resin and Buehler Epothin epoxy resins. Assessment included clarity of tissue organization and incurred thin section damage, as described previously. We found no appreciable difference in tissue clarity or visibility between specimens embedded in Buehler Epothin epoxy resin and specimens embedded in Silmar-41 polyester resin (Fig. 3). Additionally, we found no difference in thin-section quality between modern and fossil specimens regardless of resin type. None of the prepared thin-sections exhibited either type of damage prior to tissue organization being visible and identifiable under a polarized light microscope. Final thin section thicknesses ranged from 38 µm to 247 µm when tissue organization could be identified under a polarized light microscope; averaged thicknesses ranged from 46 µm to 237 µm (Table 2). Finished, averaged thin-section thickness varied between resin types despite similar tissue visibility with specimens embedded in Buehler Epothin resins ranging from 46 µm to 90 µm and specimens embedded in Silmar-41 resin ranging from 56 µm to 237 µm (see Supplemental Information 1 for individual slide thickness).

Figure 3 Tissue clarity between polyester and epoxy resin embedded specimens.

Separate parts of a fossil rib of an indeterminate taxa were embedded in (A) polyester resin and (B) epoxy resin and the finished sections imaged under linear light with a polarizing light microscope. Tissue clarity, as qualitatively assessed by the authors when viewed with a polarizing light microscope, did not appear to be affected by resin type. (C) Transverse section of the calcaneum of Dasypus novemcinctus embedded in polyester resin and (D) transverse section of Gallus gallus domesticus humerus embedded in polyester resin. Tissue clarity in modern tissue was not affected by the use of polyester resin.

Table 2 Slide thickness of each finished thin section and thickness at material loss.

Thin sections were defined as finished when tissue organization was visible and identifiable using a polarizing light microscope. Select thin sections were further ground on a lapidary wheel until material integrity was lost and the thickness of the specimen when material damage incurred was measured.

Resin type	Specimen	Element	Avg. finished slide thickness (µm)	Slide thickness
at loss (µm)	
Polyester Resin	Domestic Chicken	Right Humerus	72	–	
(Silmar-41)		Right Femur	61	–	
		Right Tibia	100	–	
	Nine-banded Armadillo	Right Calcaneum	66	60	
	Indet. Turtle	Femur	64	32	
	Gar	Scale	237	–	
	Indet. Crocodile	Scute	70	–	
	Indet. Ornithopod	Tooth	66	45	
	Unknown	Rib	56	–	
	Unknown	Fragment	80	67	
Epoxy Resin	Nine-banded Armadillo	Left Calcaneum	78	71	
(Epothin-1)	Indet. Turtle	Femur	46	44	
	Unknown	Rib	60	–	
	Indet. Ceratopsian	Tooth	81	20	
(Epothin-2)	Unknown	Fragment	90	81	

Resin type and section damage

We also did not find a difference between thin-section thickness at point of material damage, albeit with a small sample size. Table 2 lists the slide thickness at integrity loss for each specimen. Thickness at integrity loss was well beyond that in which bone microstructure was visible and identifiable in each specimen, and integrity loss resulted in damage type (1) (material tearing or popping off of the slide) (Fig. 4).

Figure 4 Example of specimen damage incurred during grinding and polishing.

(A) Thin section of fossil turtle femur embedded in polyester resin was ground on a lapidary wheel until specimen integrity was lost. (B) Inset of (A) showing region of specimen (red shade) that ripped off of the slide when ground too thin. Image taken under linear polarized light.

Discussion

Our results suggest that polyester resins can be used for embedding undecalcified modern bone and fossilized hard tissues without loss of tissue visibility or embedded material integrity. The finished section thickness did vary between Silmar-41 and Buehler Epothin embedded specimens. On average, epoxy resin embedded specimens had to be ground thinner than polyester embedded specimens to achieve similar levels of tissue visibility. However, finished section thickness appears more dependent on variation in hard tissue composition rather than resin type. For example, the fossil rib was divided into two parts and each part embedded in a different resin. The average finished section thickness of the polyester resin embedded rib part was 56 µm and the epoxy resin embedded rib part was 60 µm. Similar trends were observed in the finished slide thicknesses of the divided unknown fossil bone fragment and the two fossil teeth. The gar scale, composed of bone, dentine, and ganoin, was embedded in Silmar-41 resin and finished section thickness was 226 µm–247 µm, far thicker than any other finished thin section (Supplemental Information 1). Removal of the gar scale section thickness results reduces the polyester resin section thickness range to 38 µm–113 µm, closer in range to slides with epoxy resin embedded specimens. Ideally, a future study will embed a fossil gar scale in epoxy resin for comparison of similar specimen material compositions.

Resin type also did not appear to affect tissue visibility with the microscope or thickness at material loss during polishing. Section thickness at moment of material integrity loss was similar between resin types but varied among specimens, similar to results of finished slide thicknesses. This suggests that resin type has no appreciable effect on adherence or material loss at low section thicknesses.

Recently published methodologies for the preparation of undecalcified modern bones and small fossil hard tissues show a preference for the use of epoxy resins as embedding media (Lamm, 2013) rather than polyester-based media. In a brief survey of 134 research articles using histological sampling of either fossil bone or undecalcified modern bone (modern bone studies surveyed typically focused on non-primate tetrapods), we found polyester resins were preferably used in fossil studies (41% used a polyester resin, 32% used an epoxy resin, 26% did not use polyester or epoxy or did not specify a resin type) and epoxy resins were preferred for modern undecalcified bone (59% used an epoxy resin, 26% used a polyester resin, 15% did not use polyester or epoxy or did not specify). Epoxy resins were suggested to improve penetration and bonding of resin to the embedded hard tissue and to prevent material loss at low thin-section thickness. Polyester resins, on the other hand are recommended for larger fossil material (Wilson, 1994; Lamm, 2013), although several studies have utilized polyester resins for embedding modern undecalcified bone (e.g., Bourdon et al., 2009; Canoville, Schweitzer & Zanno, 2019). Our study suggests that the less expensive polyester resins can be used interchangeably with the more expensive epoxy resins, decreasing the costs of histological preparation. However, this is a preliminary study and other variables may affect results including selection of mounting glue, hand pressure during polishing, humidity, room temperature, silicon carbide paper quality, and lab tech experience. In addition, our study focuses on specific resins used in protocol outlined in Lamm (2013) and excludes other commonly used resins (e.g., UV curing glue, Araldite, Technovit, etc.). Our study does not examine the long-term effects of resin types in terms of color changes or changes in brittleness of embedded specimens. Lamm (2013) notes changes in glue color (yellowing) and slide peeling have occurred in a few older specimens (histologically prepared over 24 years ago) in the Museum of the Rockies histology collection. A long-term study is necessary to ensure that resin type does not have a depreciable effect on stored embedded specimens. Lastly, modern specimens sampled were collected and salvaged for the purpose of this study and chemical processing was tightly controlled by the authors. In contrast, modern specimens in museum collections may have a complex and undocumented history of chemical processing. Exposure to atypical chemicals during museum preparation and curation may have deleterious effects on embedding efficacy or long-term integrity of embedded specimens. The current study does not address any potential differences between ‘freshly collected’ modern specimens and modern specimens that have been stored long-term in museum collections, but future researchers should take into account any chemical used on specimens prior to initiating histological processing.

Conclusions

Few studies have focused on product efficacy in paleohistological methods, potentially leading to unnecessary expenses. Epoxy resins are suggested to improve resin penetration, but incur a much larger financial cost relative to polyester resins. In this preliminary study, neither tissue quality under the microscope or integrity of specimen thin sections differed between polyester and epoxy resins. Institutions processing specimens for osteohistological sampling can alleviate some financial strain by utilizing polyester resins. However, long term storage may have negative effects on one resin type more so than another. The results of this study would benefit from an increased sample size and observation of resin embed deterioration over time.

Supplemental Information

Supplemental Information 1 Raw data for finished slide thickness

Multiple thin sections were generated for some specimens. Individual section thickness is presented for each generated slide.

Click here for additional data file.

The authors would like to thank Ellen-Thérèse Lamm and Museum of the Rockies for critical discussions, materials, and extensive training in osteohistology. Thank you to Andrew Lee for suggestions on how to remove the confetti artifact from two-ton epoxy.

Additional Information and Declarations

Competing Interests

Author Contributions

Data Availability

The authors declare there are no competing interests.

Christian T. Heck conceived and designed the experiments, performed the experiments, analyzed the data, prepared figures and/or tables, authored or reviewed drafts of the paper, and approved the final draft.

Gwyneth Volkmann performed the experiments, authored or reviewed drafts of the paper, and approved the final draft.

Holly N. Woodward conceived and designed the experiments, authored or reviewed drafts of the paper, and approved the final draft.

The following information was supplied regarding data availability:

Data in the form of high quality images is available at MorphoBank: https://morphobank.org/index.php/Projects/ProjectOverview/project_id/3775.

All specimens are housed permanently at Sam Noble Museum of Natural History, Norman, OK, USA. Accession numbers: OMNH 69942, OMNH 69943, OMNH 69944, OMNH 69945, OMNH 69946, OMNH 69947, OMNH RE862, OMNH RE863.

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
