# Peer review of "Polyester or epoxy: assessing embedding product efficacy in paleohistological methods"

_PeerJ, doi:10.7717/peerj.10495_

## Round 0.1 · original submission · Minor Revisions

Thank you for your submission. This was a strong paper and I appreciate the authors work on a paleohisto techniques paper. They are too few and far between.

Please carefully review reviewer comments and provide a tracked changes copy of your revised manuscript and a rebuttal letter with your revision. In particular, I agree with reviewer 2 that an image of modern bone would enhance the paper. I am also interested if the authors could speak to vacuum impregnation (reviewer 1).

Please let me know if you have any questions.

Reviewer 1 ·

Basic reporting

no comment

Experimental design

I had several questions, all which are minor or easy to address:

1) I know it seems like a silly question, but was the chicken a roast chicken or raw? Could argue cooking affects the bone.

2) Note: Acetone can also damage the fossils, so be careful that your wash didn’t affect the mineral integrity of the fossils.

3) Curious, but would there be a difference between “fresh” extant specimens and museum-based extant specimens?

4) For the difference in penetration between the two embedding mediums, was there a correlation with which one was usually used with a vacuum? I know a number of paleohistologist who swear by vacuum impregnation.

5) Can you state what you’re desired thin section thickness range is? That way you could say “you can’t get beyond X thickness with this, but that’s OK since we only need to get to Y thickness for histological analysis”.

Validity of the findings

no comment

Additional comments

Thank you for writing a techniques papers. I feel these are much needed in paleontology.

·

Basic reporting

Overall, the manuscript by Heck and colleagues conforms to journal standards. The literature review is sufficient to give enough background to a non-specialist, and the writing is excellent. I do think that an additional figure is needed to show how polyester resin and epoxy perform for modern bone sections. Given the amount of space allotted to the preparation of modern bone, the lack of images from modern bone histology is surprising.

Experimental design

The design could use some refinement. My main suggestion relates to the variation in grinding pressure and quality of the section. In the revision, please address how grinding pressure was standardized or justify why it’s not an important factor.

I’m also curious about the use of cyanoacrylate adhesive. It’s not an ideal adhesive for applications involving high shear stress (e.g., grinding sections). Two-ton epoxy has much better shear strength. However, the “confetti” artifact is a concern. My immediate thought was contamination from the starch powder used in gloves. But that would appear as round granules with the Maltese cross artifact under crossed polarizers. My second thought was that the confetti represents the slow crystallization of epoxy. If correct, you should be able to remelt them by heating the part A component of epoxy (resin) to 50 degrees C for a few hours.

Can you discuss why specimens are allowed to air dry before embedding? Trapped air in your sample causes transmitted light to scatter away from the objective lens, in effect cloaking that part of the sample. Sections with lots of trapped air need to be ground thinner to transmit enough light for imaging. And the thinner one needs to grind the section, the more likely it will peel.

Validity of the findings

The study seeks to determine the cost-effectiveness of polyester resin when used to embed fossils for undemineralized histological sectioning. The results suggest that modern and fossil bone sections can be ground thin whether they are embedded in polyester resin or epoxy. Therefore, polyester resin, as the less expensive of the two, is more cost-effective, especially when embedding large bones. I generally agree with the conclusion and can attest to the cost-effectiveness of the polyester resin as an embedding medium (I’ve used it for nearly 30 years).

Additional comments

See annotated PDF

---

## Round 0.2 · accepted · Accept

Thank you for your submission. I think that this will be a very useful paper to bone histologists and am happy to move it forward to production. Please let me know if you have any questions.

Reviewer 1 ·

Basic reporting

no comment

Experimental design

no comment

Validity of the findings

no comment

Additional comments

Thank you for adding clarification to your methods.